# Optimal evaluation of energy yield and driving force in microbial metabolic pathway variants

**Ahmed Taha**[1], **Mauricio Patón**[1], **David R. Penas**[2], **Julio R. Banga**[2],
**Jorge Rodríguez**[1]*

**1** Department of Chemical Engineering, Research and Innovation Center on $CO_2$ and $H_2$ (RICH) Khalifa University, Abu Dhabi, United Arab Emirates, **2** Computational Biology Lab, MBG-CSIC (Spanish National Research Council), Pontevedra, Galicia, Spain

* jorge.rodriguez@ku.ac.ae

**Data Availability Statement:** All code written in support of this publication is publicly available at https://doi.org/10.5281/zenodo.7983707.

## Abstract

This work presents a methodology to evaluate the bioenergetic feasibility of alternative metabolic pathways for a given microbial conversion, optimising their energy yield and driving forces as a function of the concentration of metabolic intermediates. The tool, based on thermodynamic principles and multi-objective optimisation, accounts for pathway variants in terms of different electron carriers, as well as energy conservation (proton translocating) reactions within the pathway. The method also accommodates other constraints, some of them non-linear, such as the balance of conserved moieties. The approach involves the transformation of the maximum energy yield problem into a multi-objective mixed-integer linear optimisation problem which is then subsequently solved using the epsilon-constraint method, highlighting the trade-off between yield and rate in metabolic reactions. The methodology is applied to analyse several pathway alternatives occurring during propionate oxidation in anaerobic fermentation processes, as well as to the reverse TCA cycle pathway occurring during autotrophic microbial $CO_2$ fixation. The results obtained using the developed methodology match previously reported literature and bring about insights into the studied pathways.

## Author summary

In this study we have developed a method that evaluates the feasibility of various metabolic pathways in microbes and finds the optimum concentrations of metabolic intermediates modes in terms of energy efficiency (yield) and driving forces (rate). The method is based on thermodynamic principles and uses multi-objective optimization to evaluate numerous different pathway variants, including the use of different electron carriers and energy conservation reactions. It also considers other constraints, such as the balance of conserved components. A complex mathematical problem is solved that maps the trade-off between yield and rate in the pathways evaluated. We applied this methodology to analyse alternative pathways in propionate oxidation and $CO_2$ fixation by microbes. This research has potential applications in biotechnology and environmental studies, advancing our understanding of the trade-offs between energy efficiency and rate in different

**Funding:** AT and JR acknowledge funding by the Sustainable Bioenergy Research Consortium under the Award No. 8434000305/EX2019-003 and by resources of the Research and Innovation Center on CO2 and H2 (RICH). JRB and DRP acknowledge funding from MCIN/AEI/ 10.13039/501100011033 [project Biodynamics, ref. PID2020-117271RB-C22]. The funders had no role in study design, data collection and analysis, decision to publish, or preparation of the manuscript.

**Competing interests:** The authors have declared that no competing interests exist.

microbial pathways and allowing for the design of synthetic pathways for engineered organisms based on optimality principles.

## 1. Introduction

The laws of thermodynamics determine the ultimate limits on what metabolic pathways a living cell can drive forward. Microbial cells extract Gibbs free energy from chemical reactions with the main purpose of growth and replication [1]. In energy-limited environments, microorganisms are under high selective pressures to utilise the limited amount of energy available as efficiently as possible. This is especially true for microbes with catabolic reactions of very small driving force (where the driving force for a given reaction is the negative of the Gibbs energy dissipated by the reaction) and therefore very small amounts of Gibbs energy available. In anaerobic or in extreme environments very small driving forces can be found on which microbial growth is still possible [2]. This energy scarcity leads to a natural optimisation problem, in which the current-day existing microbes can be assumed to have approached an 'optimum solution', in terms of the most efficient use of limited resources [3,4]. Several research studies can be found that describe in detail the role of energy limitation as a driver for the evolution of more efficient mechanisms within living cells [1,5–7]. Various aspects of the highly preserved central metabolism, such as the choice of electron carriers and cofactor activation sites [8,9], have been explained through the application of such energetic constraints. In addition, some studies have explained the products of mixed culture fermentation as the result of energy conservation mechanisms within the cell such as electron bifurcation [10,11]. The possibilities unlocked upon understanding the inner workings of prokaryotic metabolism are vast; if we can optimise pathways and environmental conditions to favour certain metabolic outcomes, novel bioprocesses would become feasible. Examples of these include light-independent autotrophic $CO_2$ fixation through syngas fermentation, chain elongation to valuable medium chain solvents and fatty acids, and the biosynthesis of microbial electrofuels [12–15].

Discussion exists amongst microbiologists about whether energy yield (ATP energy recovered per mole of substrate consumed) or energy recovery rate (ATP energy recovered per unit time) is the main selection pressure for energetically constrained microbial metabolisms [6,16]. Various experimental works as well as modelling approaches have elucidated different situations in which either of the two factors dominate [2,17,18]. This question becomes more complex in mixed culture microbial ecosystems where interspecies competition for substrate and energy leads to the development of various cooperative (such as syntrophism and symbiosis) and adversarial behaviours on shared resources [16,19–21]. Although higher driving forces to achieve higher rates imply higher energy dissipation and loss of efficiency, cells can operate under high driving forces at specific stages in a pathway to create permanent irreversibility [3]. Other researchers have also asserted that the trade-off between rate and yield is not universal and that cells may seek to minimise the total enzyme mass required to perform their overall metabolism [17]. Cell metabolism is also limited by constraints other than thermodynamics including the chemical stability of metabolites at the environmental niche [3], the size of any intermediate metabolites and their crowding effect [22], or the permeability of the substrates and efficient use of limited membrane space [23].

Early research into the thermodynamic constraints on metabolism focused on overall reactions [2,24]. However, with the increased availability of computational power, researchers have moved on to analysing entire pathways (or even genome-scale metabolic networks) step by step [3,15,18,25]. Several mathematical tools and computer toolboxes for the engineering of

natural and novel metabolic pathways have emerged [26]. One very common approach is the classical Flux Balance Analysis (FBA) and its various modified forms. Such methodologies have been used to calculate steady-state fluxes that would optimise certain targets such as the yield of biomass [27]. Other researchers have improved on FBA by adding thermodynamic constraints to prevent solutions that violate the Second Law of Thermodynamics [17,25,28–30].

A common desired objective from the thermodynamic analysis component in these approaches is the maximisation of the (potentially limiting) Minimum Driving Force (MDF) among all pathway steps [3,25,30,31]. Reversible reactions running close to thermodynamic equilibrium can be modelled as squandering valuable enzyme capacity due to significant catalysis of the backwards flux. The relationship between the Gibbs energy dissipated in a reaction and its ratio of forward to backward fluxes has been used in literature as a model connecting the driving force with the reaction kinetics [3,15,25,32]. This relationship, called the flux-force efficacy (FFE) formula [3], can be presented as:

$$FFE = \frac{e^{\frac{\Delta G_{diss}}{RT}} - 1}{e^{\frac{\Delta G_{diss}}{RT}} + 1}$$

where R is the ideal gas constant (in kJ/(mol·K)) and T is the absolute temperature (in K). The value of FFE represents the fraction of enzyme active sites that are catalysing the net forward reaction with respect to the total enzyme sites catalysing the forward and backward reactions together. By increasing the driving force ($\Delta G_{diss}$), the FFE increases and thus the enzyme utilization is improved. In real terms, this translates into a faster kinetic rate using the same enzyme mass, or a similar rate using a smaller enzyme mass [3]. Previous works in literature have demonstrated that, if equal kinetic parameters are assumed for all enzymes, the maximisation of the minimum driving force (MDF) in a pathway will minimise the total enzyme mass required to carry out the conversion, thus lowering the cell's protein synthesis burden [3].

One of the most recent works on the topic of metabolic pathway analysis focused mainly on the optimisation of energy recovery via studying various possible combinations of electron carriers and membrane proton translocations in a given pathway (with each combination being called a pathway variant), taking into consideration mainly thermodynamic constraints and forgoing the flux analysis [33].

To the best of our knowledge, there has been no generalised approach that allows for the evaluation of the optimum energy yield configuration or variant of a given pathway while simultaneously manipulating the driving forces within the pathway. In this work, we present a computationally efficient methodology that achieves this through the use of established mixed-integer programming solvers. The objective of this methodology is to create a model for any desired pathway that can identify the optimum among all pathway variants, at different environmental conditions and biochemical parameter selection, in terms of both energy yield and driving force distribution. A secondary objective of the tool is to identify the thermodynamic bottlenecks, both local and distributed, that could render a pathway variant infeasible under certain environmental conditions. Such a tool could be used for generating hypotheses for experimental design as well as explaining observed experimental results, and for the design of novel biosynthesis processes.

## 2. Methods

In this section, the detailed steps of the proposed methodology for the bioenergetic evaluation of metabolic pathways are presented, from the definition of the metabolism to the formulation

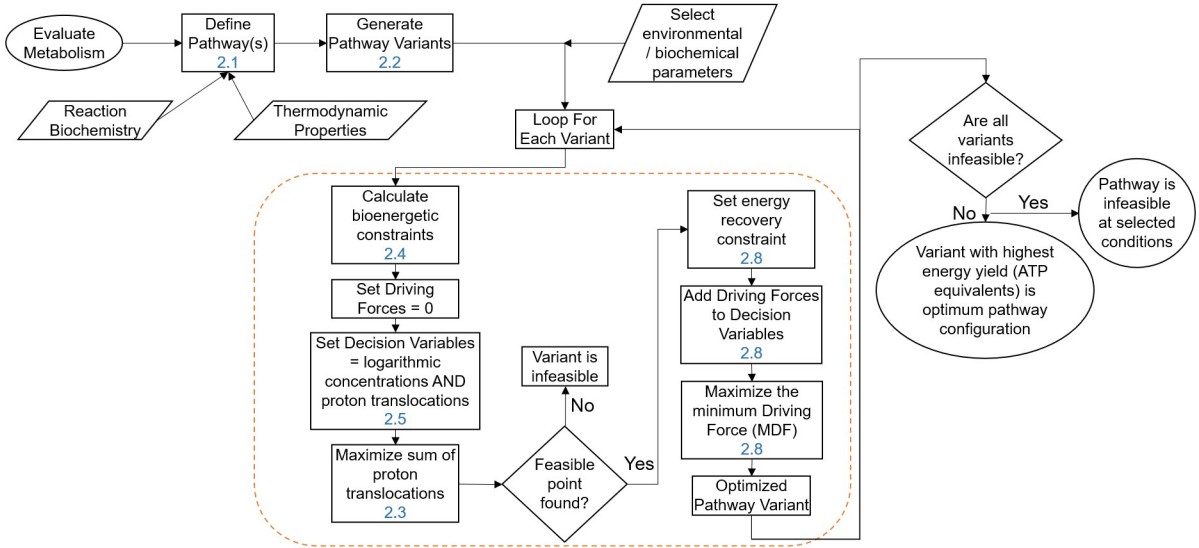

**Fig 1. Flowchart describing the steps involved in the evaluation of a given metabolic pathway.** The block within the orange dashed box constitutes the steps of one multi-objective optimisation run. The numbers in blue refer to the corresponding manuscript section with a detailed explanation.

of the optimisation problem. Fig 1 shows a summary flowchart of the method, and a detailed explanation of each step will be presented in the following subsections.

## 2.1 Metabolism, Pathway and Reaction definitions

The first step of the proposed methodology is to define the metabolic process of interest in terms of a starting point (reactant) and an endpoint (product). The metabolic pathway steps are then defined, based on known biochemistry from literature and biology databases such as KEGG and MetaCyc, as a series of biochemical reactions that connect the desired substrate and product [34,35]. For each reaction step, the reactants, products and their respective stoichiometric coefficients must be defined. For reactions that involve oxidation or reduction, the reactions are considered generically in terms of the electrons, omitting biochemistry related to possible electron carriers at this stage. It is important to ensure the overall equation of the metabolism is closed (i.e., no net production or consumption of any intermediate metabolites).

## 2.2 Generation of pathway variants

For this work, a pathway variant is defined as a set of reaction steps from start to end for which all its reactions are balanced and all redox reactions are allocated to a biochemically possible electron carrier (e.g., ferredoxin, NADH, $FADH_2$). The permissible electron carriers for a given reaction are identified from biology databases such as KEGG and MetaCyc. In addition, only feasible regeneration reactions of the carriers involved are included in the pathway variant. For example, if the terminal electron donor for a reductive pathway is hydrogen and the pathway variant uses $FADH_2$ as an electron donor for at least one intermediate reaction, it is not possible to re-reduce the resultant FAD directly against hydrogen as there is no known enzyme that catalyses this reaction. Instead, in this example, the reaction of FAD with NADH must be included in the pathway as it is catalysed by a well-documented oxidoreductase [36], and the reaction of NAD with hydrogen will also be included as the terminal carrier

regeneration reaction. This ensures adherence to known biochemistry to allow for a reaction to be included.

The total number of pathway variants possible results from the product of the number of permissible electron carriers for each redox reaction for the entire pathway.

## 2.3 Optimisation objectives

Once a set of metabolic pathways and/or their variants are appropriately defined, the aim is to find the pathway variant that is most favourable in bioenergetic terms, i.e., that configuration in terms of intermediate concentrations that yields the largest energy recovery in terms of net ATP/proton translocations. The methodology can combine two objectives, namely the maximisation of the recovered energy (ATP yield) of the metabolic pathway as well as the driving force maximisation for the minimum driving force step (MDF) as defined in the introduction. This objective is considered of special relevance for microorganisms living in energy-limited environments [18,37]. A multi-objective optimisation formulation is therefore adopted for each pathway variant evaluated, with the following objectives:

$$\max_{\underline{x}} \ \underline{J} = \begin{bmatrix} \sum n_{p,j} \\ B \end{bmatrix}$$

subject to the constraints that will be defined in subsections 2.4–2.6, where the first term of the objective vector $\underline{J}$ is the sum of proton translocations and $B$ is the minimum driving force (MDF), and $\underline{x}$ is the vector of decision variables that will be defined in subsection 2.3. Maximising the sum of the proton translocations maximises the bioenergetic yield of the pathway, while the maximisation of the MDF allows for a faster kinetic rate, or a similar rate using a smaller enzyme mass [3]. The above optimisation problem will be solved by the epsilon-constraint method [38,39].

Other important assumptions under this approach include:

1. All reactions for a given pathway are assumed to occur within a single cell compartment. No further compartmentalization is considered in the model; thus, no intracellular transport reactions are included in the analysis.

2. The activity of all species is taken to be equal to their concentrations i.e., all activity coefficients equal one. This assumption can be waived with the inclusion of an appropriate activity model such as the Debye-Hückel law, though these calculations are frequently imperfect in biological systems due to crowding effects [40,41].

3. The only energy-recovering and conserving mechanisms within the cell are substrate-level phosphorylation (SLP) and chemiosmotic proton translocations [42].

4. The model is steady-state and time-independent. No changes to concentrations of metabolites or biochemical parameters are part of the optimisation.

## 2.4 Definition of reaction energetics

**2.4.1 Chemical potentials.**   In order to define the problem constraints, the terms that make up the Gibbs energy expression for a given reaction are defined. The standard Gibbs (in kJ/mol) of formation in aqueous solution ($\Delta G_f^0$) for all species involved in the pathway are obtained from biochemical databases such as [43,44] or via empirical group contribution methods such as the ones developed by [45,46]. The effect of temperature is accounted for

through the Van't Hoff equation is applied as follows:

$$\Delta G_{f,i}{}^{0}(T) = \Delta G_{f,i}{}^{0}\left(T_{ref}\right) * \frac{T}{T_{ref}} + \Delta H_{f,i}{}^{0} * \left(1 - \frac{T}{T_{ref}}\right) \tag{1}$$

where $\Delta G_{f,i}{}^{0}(T)$ is the Gibbs of formation for species i at the desired temperature, $\Delta G_{f,i}{}^{0}(T_{ref})$ is the standard Gibbs of formation for species i at the reference temperature (usually 298.15 K) and $\Delta H_{f,i}{}^{0}$ is the standard enthalpy of formation for species i (in kJ/mol). The required formation enthalpies are usually harder to find in literature, so if temperature sensitivity is crucial, they could also be estimated using group contribution methods [45,47].

The chemical potential of a species i ($\mu_i$) is then calculated using its formation Gibbs as well as its activity according to:

$$\mu_i = \Delta G_{f,i}{}^{0} + RT\ln(\gamma_i C_i) \tag{2}$$

where R is the ideal gas constant (in kJ/(mol·K)), $C_i$ is the concentration of species i (in mol/L) and $\gamma_i$ is the activity coefficient of species i (here assumed to be 1 as discussed in section 2.3).

**2.4.2 Substrate-level phosphorylation.**   Substrate-level phosphorylation (SLP) involves the direct hydrolysis of the phosphoanhydride bond(s) in ATP to produce ADP or AMP, or the reverse reaction. To account for SLP in this model, we define the energy associated with the hydrolysis of ATP into ADP and inorganic phosphate to be one 'unit', $\Delta G_{ATP}$. The value of $\Delta G_{ATP}$ varies depending on cellular conditions but is typically given in the literature to be between -45 to -70 kJ/mol [33,48,49]. Reactions involving AMP are also biologically relevant and the relative energy associated with the hydrolysis of ATP to AMP and pyrophosphate ($PP_i$) is roughly 40–50% higher than $\Delta G_{ATP}$ [50,51].

Information about SLP is tabulated for all reactions in the pathway of interest from biochemical databases. As the participation of ATP is always accompanied by ADP and/or AMP, they can be represented as an energy 'token', thus removing these species from consideration and instead introducing an energy factor denoted as L. For a reaction j, the sign of $L_j$ is given by whether energy is invested (ATP consumed) or recovered (ATP formed), and is taken to be positive for recovery. For ATP/ADP reactions, the magnitude of $L_j$ is taken to be 1. For ATP/AMP reactions, the magnitude is taken to be 1.5. For a reaction j where SLP cannot occur, $L_j$ is necessarily zero.

**2.4.3 Proton translocations.**   Reactions catalysed by membrane-bound enzymes are assumed to have the potential to conserve energy via protons translocation across the membrane [42]. This can chemiosmotically help drive a reaction (by moving the protons down their electrochemical potential gradient) or recover energy from a sufficiently exergonic reaction (by moving the protons against their gradient). The energy associated with a proton translocation, named the proton motive force (*pmf*), is the smallest quantum of energy biological systems can work with, and its magnitude is the sum of both the chemical potential and electric potential gradients. The *pmf* value is frequently considered in relation to the ATP hydrolysis energy, and a typical value for this ratio of proton translocations needed per ATP molecule synthesized (henceforth referred to as $r_{H/ATP}$) is 9/3, although a range of values is reported in the literature [33,52]. For the simulations conducted in this work, a value of 10/3 was used.

The cellular location of each reaction in the pathway and its corresponding enzyme is obtained from biochemical databases in order to define all the possible chemiosmotic energy conservation sites. For a given reaction j allowing for proton translocations, a new energy term

for the reaction is introduced as follows:

$$\Delta G_{H^+,j} = n_{p,j} * \frac{\Delta G_{ATP}}{r_{\frac{H^+}{ATP}}}$$
(3)

where $n_{p,j}$ is an integer number of proton translocations that occur in reaction j. Like with SLP, the sign convention is that $n_{p,j}$ is positive for proton translocations recovered (protons pumped against their gradient), negative for proton translocations invested (protons moved down their gradient) and zero for reactions that do not translocate protons.

**2.4.4 Gibbs energy of elementary reactions.** The overall change in Gibbs energy associated with a reaction is the most central concept in this framework and can be calculated by summing up its constituents discussed so far, with chemical potentials weighted by their stoichiometric coefficients in the reaction. For a reaction j, the change in Gibbs energy $\Delta G_r$ is given by:

$$\Delta G_r = \sum \left( v_i * \mu_i \right) + L_j \Delta G_{ATP} + n_{p,j} \frac{\Delta G_{ATP}}{r_{\frac{H^+}{ATP}}} \quad n_{p,j} \in \mathbb{Z}$$
(4)

where $v_i$ is the stoichiometric coefficient of species i involved in the reaction, $\mu_i$ is the chemical potential of species i, L is the number of ATP energy tokens involved in the reaction and $n_{p,j}$ is the integer number of proton translocations in the reaction. By expanding the definition of chemical potential ($\mu_i$), we arrive at the following form for $\Delta G_r$:

$$\Delta G_r = \sum \left( v_i * \Delta G_{f,i}^0 \right) + RT \sum \left( v_i \ln C_i \right) + L_j \Delta G_{ATP} + n_{p,j} \frac{\Delta G_{ATP}}{r_{\frac{H^+}{ATP}}} n_{p,j} \in \mathbb{Z}$$
(5)

The first term on the right-hand side of the above equation is usually called the standard Gibbs energy change of the reaction, $\Delta G^0$.

## 2.5 Selection of decision variables

To proceed with the optimisation formulation of the problem, the decision variables need to be identified. By inspecting Eq (5), it is directly inferred that the Gibbs energies depend on the concentrations of species ($C_j$) involved in the pathway variant as well as the number of possible proton translocations ($n_{p,j}$) in all the individual reactions. In order to preserve the linearity of Eq (5), the natural logarithm of the concentration is used as the decision variable instead of the molar concentration. Thus, the vector of decision variables results into:

$$\underline{x} = \begin{bmatrix} lnC_1 \\ \vdots \\ lnC_n \\ n_{p,1} \\ \vdots \\ n_{p,j} \end{bmatrix}$$
(6)

Some species whose concentration is determined externally to the cell as well as some biochemical parameters are excluded from the decision variables vector $\underline{x}$, specifically:

1. The external concentration of reactants and products is defined by the environmental conditions experienced by the organisms. This includes pH values, the external substrate **S** and other cofactors such as free coenzyme A. Depending on the model objectives, some of these

concentrations can be either fixed or made into decision variables or parameters to be evaluated.

2. The concentrations of certain so-called conserved moieties may be prefixed to values found in the literature. This is usually applicable to common electron carrier concentrations such as NAD and ferredoxin which participate in hundreds of reactions across the entire metabolism of the cell.

Since the decision variables include logarithmic concentrations, which are continuous variables, and the number of proton translocations, which are integer variables, the problem becomes a mixed-integer linear programming (MILP) problem.

## 2.6 Energetic constraints

**2.6.1 Definition.**   The second law of thermodynamics imposes an energetic constraint to all individual reaction steps in any metabolic pathway; the change in total Gibbs ($\Delta G_r$) energy must be negative for any reaction step to proceed forwards.

For a reaction at equilibrium, the value of $\Delta G_r$ is zero, and the forward and backward rates of the reaction are equal. When $\Delta G_r$ is negative, that energy is dissipated and is usually referred to as the reaction's thermodynamic driving force [3].

A threshold minimum Gibbs energy change of the reaction ($\Delta G_{min}$) is necessary for any reaction to proceed at a finite sensible rate and can be imposed as a non-zero value.

$$\Delta G_r - \Delta G_{min} \leq 0 \tag{7}$$

**2.6.2 Matrix formulation.**   In order to write the thermodynamic constraints as a matrix for linear programming, we combine Eqs (5) and (7) to get the constraint for a reaction j as follows:

$$\Delta G_j^0 + RT \sum \left( v_{i,j} \ln C_i \right) + L_j \Delta G_{ATP} + n_{p,j} \frac{\Delta G_{ATP}}{r_{\frac{H^+}{ATP}}} - \Delta G_{min,j} \leq 0 \tag{8}$$

Dividing the above equation by RT and moving terms that do not depend on the decision variables to the right-hand side allows us to rewrite it as:

$$\sum \left( v_{i,j} \ln C_i \right) + \frac{\Delta G_{ATP}}{r_{H^+/_{ATP}} RT} n_{p,j} \leq - \frac{\Delta G_j^0 + L_j \Delta G_{ATP}}{RT} - \frac{\Delta G_{min,j}}{RT} \tag{9}$$

From Eq (9), we define the *pmf* to be the coefficient of $n_{p,j}$, and the driving force term for the reaction $F_j$ to be the last term on the right-hand side. By observation, we identify the first term on the right-hand side to be the logarithm of the equilibrium constant for the reaction, $K_j$. The value of *pmf* is common to all reactions, while $F_j$ can be given a separate value for every reaction. The final equation for a single reaction constraint is thus:

$$\sum \left( v_{i,j} \ln C_i \right) + pmf * n_{p,j} \leq ln\, K_j + F_j \tag{10}$$

Note that the left-hand side of the equation is a linear combination of decision variables. By writing Eq (11) for every reaction in the pathway variant, we can assemble a matrix equation

as follows:

$$\underline{S} * \begin{bmatrix} ln\ C_1 \\ ln\ C_2 \\ \vdots \\ ln\ C_n \end{bmatrix} + pmf * \begin{bmatrix} n_{p,1} \\ n_{p,2} \\ \vdots \\ n_{p,j} \end{bmatrix} \leq \begin{bmatrix} F_1 + ln\ K_1 \\ F_2 + ln\ K_2 \\ \vdots \\ F_j + ln\ K_j \end{bmatrix} \tag{11}$$

where $\underline{S}$ is the stoichiometry matrix where columns represent reactions and columns represent species. Lastly, the two terms on the left-hand side can be combined into a singular matrix-vector product where the vector is the decision variable vector $\underline{x}$ defined in section 2.3. The overall equation will be:

$$\underline{A} * \underline{x} \leq \underline{b} \tag{12}$$

where the rows of $\underline{b}$ are the sum of the driving force term and equilibrium constant term for each reaction, and $\underline{A}$ is a concatenated matrix of $\underline{S}$ and $pmf$ in the appropriate indices. By inspecting the form of Eq (13), we conclude that the canonical form of MILP constraints has been obtained.

## 2.7 Decision Variable Boundaries

Defining appropriate boundaries for the decision variable vector $\underline{x}$ is important for efficient optimisation. The concentrations of all metabolites involved are bounded by physiologically acceptable limits based on reported biochemistry literature and kinetic considerations [53,54]. The selected upper bound for metabolite concentration is constrained by tonicity considerations, as having higher concentrations lowers the cell's osmotic potential. A typical upper bound value is $10^{-2}$ mol/L. The selected lower bound is constrained by a ballpark estimation of the cellular volumes (around $1\mu m^3$) and the number of molecules of a given species in the entire cell at low concentration. A typical lower bound value is $10^{-6}$ mol/L. Since the natural logarithm is a monotonically increasing function over its entire domain, it can be applied to the concentration inequality without loss of generality.

$$ln\ C_{min} \leq ln\ C_i \leq ln\ C_{max} \tag{13}$$

Next, we consider the appropriate bounds for the number of proton translocations in a given reaction. While no specific theoretical limit to such a process was found in literature,, we base our boundary on the fact that chemiosmotic mechanisms are a way to utilise smaller quanta of energy than SLP. As such, using the value of $r_{H/ATP}$ (rounded down to the nearest integer) in either direction was taken as the boundary. Leaving the maximum number of proton translocations permitted in a single step unbounded is also a valid option, though it can significantly increase the computational time needed depending on the solver. Note that while these boundaries limit the individual reactions' proton translocations, the sum of proton translocations in all the pathway variant's reactions is implicitly constrained by the total amount of available energy in the pathway. Therefore, there is no concern about having an unbounded optimisation problem in terms of proton translocation.

## 2.8 Additional constraints

In addition to thermodynamic constraints, additional restrictions of biochemical nature can be imposed on the system. Among these, constraints are applied to so-called conserved moieties, groups of atoms that swap between usually two forms but are not net generated or consumed by the metabolic reactions and are instead transferred between substrates. Examples of

these include the coenzyme A group species (CoA), and inorganic phosphate (Pi) both of which are frequently used to 'activate' specific substrates, and also electron carriers. The assumption that the total pool of a conserved moiety is bounded by the limits described in section 2.7 appears more reasonable than individual bounds applying individually to each species of that moiety. To capture this, a new constraint in Eq (15) is introduced based on a mass balance:

$$C_{min} \leq \sum C_{i,CM} \leq C_{max} \tag{14}$$

where the summation is applied to every species $C_{i,CM}$ containing the conserved moiety in question. Eq (15) implicitly defines two additional inequality constraints to be added to the system. In this model, we recommend the summation to be bounded by inequality constrains rather than strict equality constrains as we believe an equality constrain is too restrictive, and that the cell can choose to synthesize more of the free moiety or breakdown excess depending on the energetic situation.

Unfortunately, the additional constraints are non-linear with respect to the decision variables, and choosing to include these constraints transforms the problem into a mixed integer non-linear program (MINLP). Since this can come at a significant cost to computation time and even global optimality assurances depending on the choice of the solver and length of the pathways being studied [55], therefore its inclusion or not must be carefully considered.

To express the constraints from Eq (15) in terms of the decision variables, the exponential function is used:

$$C_{min} \leq \sum \exp(lnC_{i,CM}) \leq C_{max} \tag{15}$$

## 2.9 Solution strategy

Among the several possible approaches to solve the multi-objective mixed-integer optimisation problem, the epsilon-constraint method was used, which allows for the solution to be obtained by only two sequential optimisation runs.

Since ATP generation via SLP is stoichiometrically fixed, the only possible maximisation of energy recovery is via the net number of proton translocations. Thus, the first optimisation run maximises energy recovery only as the sum of proton net translocations across all reaction steps in a given pathway variant. During this first optimisation run all the driving force terms in Eq (12) are set to zero.

For the cases where a feasible optimum is found, this constitutes a baseline for the best possible energy yield for the pathway since no driving force forced energy dissipation was required. It is then the focus of the second optimisation run to maximise the magnitude of the smallest driving forces (considered to be the potentially kinetically limiting steps) while retaining the same energy yield, or alternatively allowing for the concession of a number of proton translocations (energy units) as a trade-off with driving force (kinetic performance). In this second optimisation run the first objective function is now a linear constraint:

$$\sum n_{p,j} = n_{p,opt} - \varepsilon \tag{16}$$

where $n_{p,opt}$ is the optimum sum of proton translocations found in the first optimisation run, and $\varepsilon$ is the (user-defined parameter) indicating how many (if any) proton translocations are permitted to be lost. This parameter $\varepsilon$ acts effectively as a weighting between the two objectives of energy and rate being analysed, i.e., the higher the number of energy units conceded ($\varepsilon$) is, the more importance to driving forces is given at the expense of energy yield.

**Table 1. A summary of the multi-objective optimisation components presented in section 2 and the epsilon-constraint solution strategy.**

| Component | First Optimisation | Second Optimisation |
|---|---|---|
| Objective | max $\sum n_{p,j}$ | max $B$ |
| Decision Variables | $\ln C_i$, $n_{p,j}$ | $\ln C_i$, $n_{p,j}$, $F_j$, $B$ |
| Thermodynamic Constraints | $\underline{S} * \begin{bmatrix} ln\ C_1 \\ ln\ C_2 \\ \vdots \\ ln\ C_n \end{bmatrix} + pmf * \begin{bmatrix} n_{p,1} \\ n_{p,2} \\ \vdots \\ n_{p,j} \end{bmatrix} \leq \begin{bmatrix} F_1 + ln\ K_1 \\ F_2 + ln\ K_2 \\ \vdots \\ F_j + ln\ K_j \end{bmatrix}$ | $\underline{S} * \begin{bmatrix} ln\ C_1 \\ ln\ C_2 \\ \vdots \\ ln\ C_n \end{bmatrix} + pmf * \begin{bmatrix} n_{p,1} \\ n_{p,2} \\ \vdots \\ n_{p,j} \end{bmatrix} + \begin{bmatrix} F_1 \\ F_2 \\ \vdots \\ F_j \end{bmatrix} \leq \begin{bmatrix} ln\ K_1 \\ ln\ K_2 \\ \vdots \\ ln\ K_j \end{bmatrix}$ |
| Additional Constraints | Conserved Moieties: $C_{min} \leq \sum \exp(lnC_{i,CM}) \leq C_{max}$ | First objective optimality: $\sum n_{p,j} = n_{p,opt} - \varepsilon$ <br> Conserved Moieties: $C_{min} \leq \sum \exp(lnC_{i,CM}) \leq C_{max}$ |
| Boundaries | $ln\ C_{min} \leq ln\ C_i \leq ln\ C_{max}$ <br> $n_{p,min} \leq n_{p,j} \leq n_{p,max}$ | $ln\ C_{min} \leq ln\ C_i \leq ln\ C_{max}$ <br> $n_{p,min} \leq n_{p,j} \leq n_{p,max}$ <br> $F_j \geq B$ |

The objective function of the second optimisation run seeks to maximise the smallest of the driving forces $F_j$ in all reaction steps in a pathway variant (i.e., the potential rate-limiting step). This objective is linearly stated using the minimax approach typically in classic optimisation textbooks; namely, a new decision variable B is defined and additional boundaries are added requiring each individual $F_j$ to be greater than or equal to B [56]. The objective is then to maximise B, which implicitly forces all the individual driving forces to be maximised as well.

The definitions of the two sequential optimisation runs are summarised in Table 1.

## 2.10 Software implementation and solvers

The methodology developed was implemented into a multi-platform toolbox; first, the biochemical components including the species and reaction data are defined in a Microsoft Excel spreadsheet, then a MATLAB script reads the data and defines the pathway variants as well as performs other necessary intermediate calculations. Finally, MATLAB calls one of three optimisation solvers implemented for this work based on user choice. The first solver implemented is the native MATLAB *intlinprog* solver, which is highly deterministic and offers high speed. However, it is not capable of accepting non-linear constraints. The second solver implemented is MEIGO, a mixed integer optimisation tool developed specifically for use in biological systems [55]. It can handle non-linear constraints as well as reliably reach a global optimum for problems with hundreds of decision variables. However, due to the random search nature of its algorithm, its computation speed is noticeably slower. The final solver implemented for this work is Gurobi, a specialized commercial tool for solving optimisation problems [57]. Gurobi handles non-linear constraints and features solution speeds superior to even the native MATLAB solver. However, it is closed-source and requires purchasing a license for non-academic use. The choice of solver did not have any impact on the results presented in section 3, and this agreement in results could be interpreted as a sign that global optimality was likely. A summary of the advantages of each solver is presented in Table 2. All code written in support of this publication is publicly available at https://doi.org/10.5281/zenodo.7983707.

## 3. Results and Discussion

The methodology was applied to two example microbial conversion pathways of interest, one of catabolic and one of anabolic nature. This section presents the results obtained and discusses conclusions that can be inferred.

**Table 2. Features of the solvers used to perform the metabolic pathway optimisations.**

| Solver | intlinprog | MEIGO | Gurobi |
|---|---|---|---|
| Documentation | [58] | [55] | [57] |
| Algorithm | Branch and Bound | Enhanced Scatter Search | Proprietary algorithm |
| License | Commercial/Academic | GPLv3 | Commercial/Academic |
| Relative Time | 1 | 2 | 0.8 |
| Non-linear Constraints | No | Yes | Yes |

## 3.1 Case Study: Catabolic propionate oxidation

The oxidation of propionate to acetate is one of several volatile fatty acid (VFA) oxidation metabolisms that occur in mixed-culture fermentations. The typical balanced equation for this metabolic process is:

$$C_2H_5COO^- + 2H_2O \rightarrow CH_3COO^- + CO_2 + 3H_2 \ \Delta G^0 = +133 kJ/mol \tag{17}$$

While the reaction is endergonic under standard conditions, the typical low environmental concentrations of hydrogen (1 to 10 nmol/L) [33,48] allow the reaction to be slightly exergonic and capable of supporting microbial life. The ability of propionate oxidizers to proceed near thermodynamic limits has been widely studied in the literature, with their feasibility being contingent on very low hydrogen partial pressures [59,60]. Previous works have compiled and expounded in detail the various pathways available for this catabolic process [61,62], and a complete domain search was performed to identify the most feasible pathway variants at different environmental conditions [33]. The multi-objective optimisation approach described in this work is applied to all the possible pathways, summarized visually in Fig 2. Complete details about the pathways evaluated are presented in Table A in S1 Text.

**3.1.1 Selected results and discussion.** The performance of the different pathways evaluated, in terms of their ATP yield and maximum MDF, is presented in Table 3 for a specific set of environmental and biochemical parameters. A preliminary simulation was performed with the exact same set of parameters and conditions as in [33] where a comprehensive bioenergetic analysis of propionate oxidation pathways was performed., and the results matched exactly those reported in that work, which confirms global optimality in this case, and serves as a strong validator of the methodology implementation.

In our analysis, the range of permissible proton translocations within a single energy-conserving step is expanded beyond that in [33], taking advantage of the computational efficiency of this method with respect to [33]. This resulted in each pathway generally having a higher energy yield. It can be observed that the optimum pathways in terms of ATP yield, are the lactate and hydroxypropionyl pathways, yielding the equivalent of 1.0 moles of ATP per mole of propionate oxidized. Note that in this particular case, the pathway variants are equally optimal, suggesting that there is no energetic advantage in using any specific combination of electron carriers under these circumstances.

The yield efficiency for the three most efficient pathways is 0.55, indicating that about 55% of the energy in this metabolism can be recovered as useable ATP, and the remaining 45% are dissipated as driving forces for the individual reactions within the pathways. Of these three pathways, the lactate pathway yields the maximum MDF value of about -1.5 kJ/mol. Using the flux-force efficacy formula presented in section 1, this is the equivalent of a 28.5% flux-force efficacy [3]. In other words, 64.2% of the enzyme molecules of the rate-limiting reaction(s) are

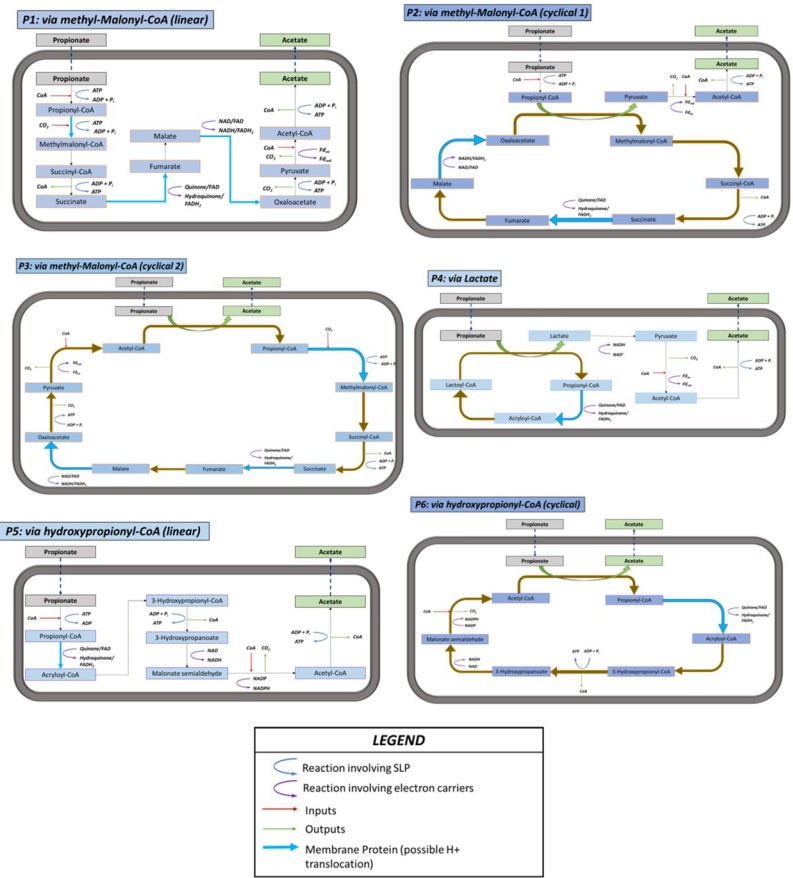

**Fig 2. Representation of the six pathways selected for propionate oxidation.**

being used to catalyse the forward flux, and 35.8% of them are catalysing the backward flux. The magnitude of these figures needs to be interpreted in conjunction with both the enzyme kinetic parameters and the relative importance of the bioenergetic recovery rate to propionate oxidizers as compared to the yield.

**Table 3. Performance of propionate oxidation pathways at $C_{Pro^-}$ = 0.01 mol/L, $C_{H2}$ = 1 nmol/L, $pH_{in}$ = $pH_{out}$ = 7, T = 35°C, $\Delta G_{ATP}$ = 50 kJ/mol, $r_{H/ATP}$ = 10/3, free CoA = 0.001 mol/L, total CoA pool < 0.01 mol/L.** The available Gibbs energy for the overall reaction (Eq 17) is 63.3 kJ/mol under these conditions ($\Delta G_{overall}$ = -63.3 kJ/mol).

| Pathway | Maximum Net ATP Yield (mol $ATP_{recovered}$ / $mol_{Pro^-}$) | Maximum Net Energy Yield (kJ$_{recovered}$/$mol_{Pro^-}$) | Energy recovery efficiency (% kJ$_{recovered}$ / kJ$_{available}$) | Maximum MDF (kJ/mol) |
|---|---|---|---|---|
| Via Methylmalonyl CoA Linear | 0.4 | 20.0 | 31.6 | 0.57 |
| Via Methylmalonyl CoA Cyclical 1 | 0.7 | 35.0 | 55.3 | 0.18 |
| Via Methylmalonyl CoA Cyclical 2 | 0.7 | 35.0 | 55.3 | 0.19 |
| Via Lactate | 1.0 | 50.0 | 79.0 | 1.10 |
| Via Hydroxypropionyl CoA Linear | 1.0 | 50.0 | 79.0 | 1.02 |
| Via Hydroxypropionyl CoA Cyclical | 1.0 | 50.0 | 79.0 | 1.10 |

The value of the maximum MDF of a pathway, if equal for all reaction steps, can be calculated by dividing the total dissipated energy (difference between total Gibbs energy available and energy recovered) by the number of reactions in the pathway. Such equal distribution of driving force throughout the pathway steps results in the minimum total enzyme mass required for the pathway. This even distribution of driving forces is however not always possible to be achieved due to bottlenecks, and the maximum MDF is limited by a single reaction while the excess Gibbs can be dissipated in various reactions. This implies that, although an optimum value can be found that is unique in terms of energy recovery and MDF, the set of decision variable values to achieve that optimum is not, leaving some slack for variations in those concentrations leading to the same optimum.

The trade-off between the energy recovery yield (represented by net proton translocations) and kinetic rate (represented by MDF) can be studied by sequentially altering the value of epsilon in Eq (16), the weighting coefficient between the two objectives of the methodology (i.e., allowing for the trade of protons recovered for a larger driving force). Fig 3 shows the Pareto front curve emerging from the analysis. Each data point corresponds to the most optimal pathway variant at the given set of conditions. The Pareto fronts indicate how, for the set of environmental conditions evaluated, specific pathways (namely via lactate and via hydroxypropionyl CoA) are the most optimal regardless of the energy recovery yield. The rightmost end of each curve line represents the absolute limit of energy recovery for each pathway at the smallest possible driving force likely leading to very low rates. Priority to energy recovery versus driving force is expected to occur in scenarios of substrate limitation where substrate full utilization would lead to maximum growth rate since the turnover rate is already limited. In the region of maximum energy recovery, several pathways appear to converge to similar optimum points. However, as we move towards higher driving forces (sacrificing proton translocations for improved MDF values), the pathways diverge. This indicates differences in how pathways are restrained by bottlenecks, with some able to transition more evenly in their utilization of the available Gibbs energy amongst its reaction steps.

For a cell, regular operation in the negative or zero energy recovery region would not be able to sustain growth as it would be spending instead of recovering energy from this catabolic

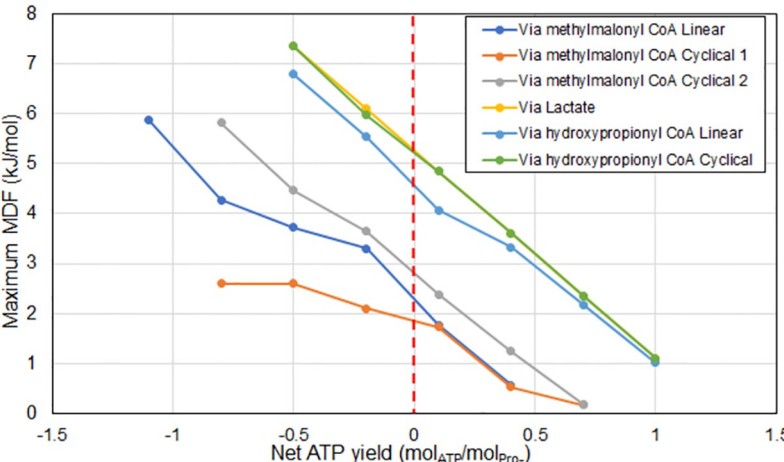

**Fig 3. Pareto curves for propionate oxidation pathways.** The pathways were evaluated at the same parameters as Table 3. The horizontal axis represents the net energy recovered from the pathway, and the vertical axis represents the maximum MDF possible for the pathway with that energy recovery. The various points are obtained by changing the value of ε (number of proton translocations sacrificed) from 0 to 5.

process. While sustained growth is indeed impossible in such a scenario, a cell may choose to occasionally utilize the pathway under those conditions for secondary reasons such as the elimination of excess propionate (reaching toxicity levels) or recycling of electron carriers (such as NADH).

Fig 4 shows a detailed analysis of the most favourable pathway (via lactate) in terms of intermediate metabolite concentration and energy recovery points. It is shown how the oxidation of propionyl-CoA to acryloyl-CoA is the main bottleneck in the pathway; not only does it require an investment of energy in the form of a proton translocation but also the product concentration to be at the lowest permissible and the reactant at almost the highest in order to proceed with the needed driving force ($\Delta G$ = -1.49 kJ/mol). In this particular pathway the distribution of proton translocation locations for this optimum total energy recovery happens to be not unique (see Fig A in S1 Text); solving the problem from different starting points, several different arrangements of the proton translocation locations are shown to lead to the same optimum in terms of both the net proton translocation and maximum MDF. The method developed allows for the observation of this flexibility in some steps while not in others and brings insight for a better understanding of this and other pathways evaluated.

### 3.2 Case Study: Anabolic $CO_2$ fixation via reverse TCA cycle

In a second case study an anabolic pathway, also known to operate close to the thermodynamic limit, is evaluated using the methodology developed. The reverse tricarboxylic acid (TCA) cycle pathway is known to be used by several autotrophic bacteria for $CO_2$ fixation [15]. The balanced complete chemical equation for the pathway (using hydrogen as an electron donor) is:

$$2CO_2 + 4H_2 + CoA \rightarrow AcCoA + 3H_2O \; \Delta G^0 = -142 \, kJ/mol \tag{18}$$

Many of the reactions in this pathway are part of the well-characterized cyclic component of aerobic respiration, but proceeding in the reverse direction. While there are some reports of aerobic bacteria strains growing on this pathway [63], the pathway tends to be anaerobic due to the oxygen sensitivity of some of its enzymes such as pyruvate synthase [64]. The absence of oxygen (and other strong electron acceptors) together with the typically low concentration of $CO_2$ normally found in natural environments, implies that the microorganisms operate under

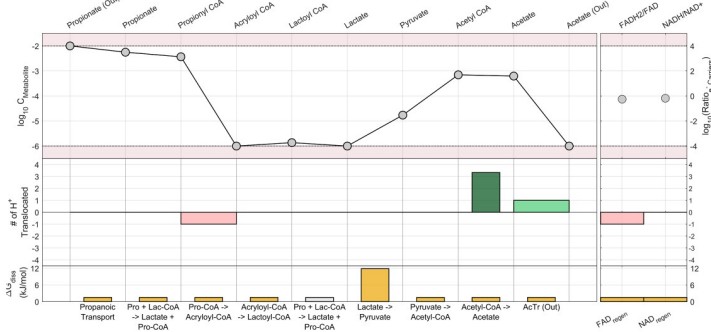

**Fig 4. Detailed bioenergetic breakdown of the via lactate pathway, evaluated at the same parameters as listed in Table 3.** The wide red bars represent proton translocations invested and the bright green bars represent proton translocations recovered. The narrow dark green bars represent SLP. A greyed-out bar indicates a reaction that's repeated for the sequential listing of species. The species are arranged in the order they react and are produced by the pathway, allowing for visual identification of bottlenecks.

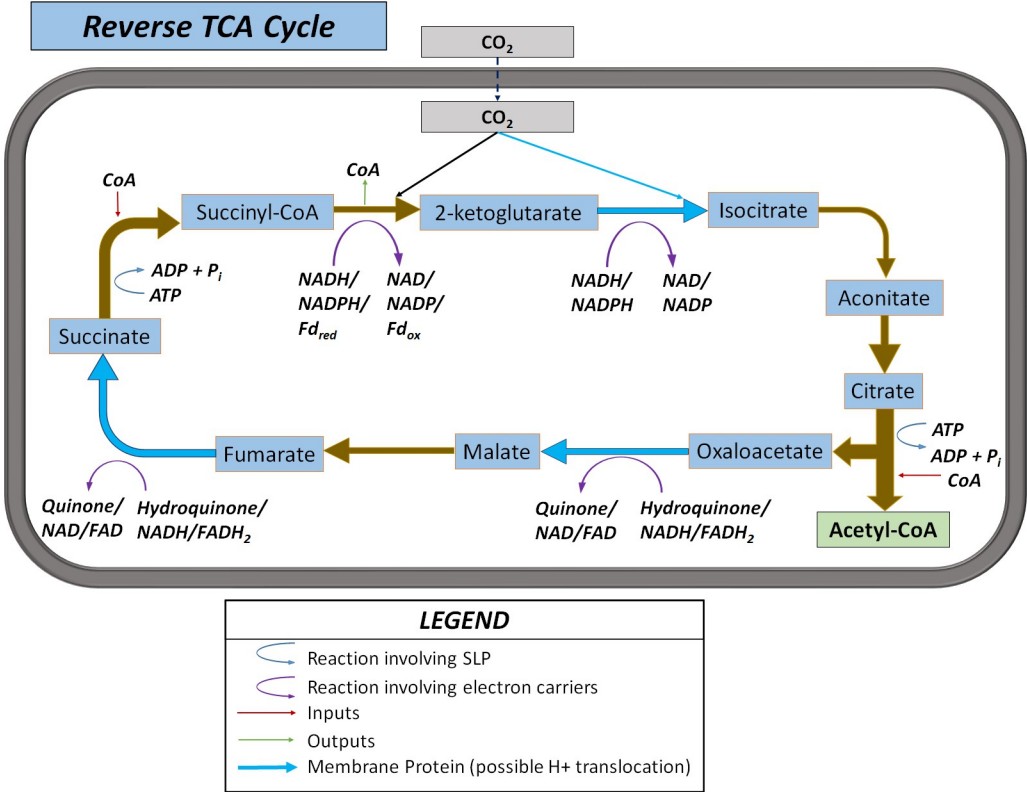

**Fig 5. Anabolic $CO_2$ fixation pathway via reverse TCA cycle.**

energy-scarce conditions and smaller amounts of energy are invested (two molecules of ATP spent through SLP per one molecule of acetyl CoA synthesized) compared to other prokaryotic $CO_2$ fixing pathways. A visual summary of the pathway is presented in Fig 5. All details about the reactions in this pathway used for the simulations are presented in Table B in S1 Text.

**3.2.1 Selected results and discussion.** The performance of the reverse TCA cycle, as measured by its ATP yield and maximum MDF, at a typical set of environmental and biochemical parameters is presented in Table 4. The pathway requires energy expenditure at the selected typical environmental conditions, except above a high inorganic carbon concentration threshold ($C_{ic}$). The pathway has 54 variants including all possible electron carrier combinations, of which 46 are equally optimal in terms of minimum energy expenditure. The actual energy required depends strongly on $C_{ic}$ and is shown to range from -0.1 to 0.8 mol ATP spent per

**Table 4. Performance of reverse TCA cycle pathway at selected concentrations of inorganic carbon, $pH_{in} = pH_{out} = 7$, $T = 25°C$, $\Delta G_{ATP} = 50$ kJ/mol, $r_{H/ATP} = 10/3$, $C_{H2} = 3$ μmol/L, free CoA = 0.001 mol/L, total CoA pool $< 0.01$ mol/L.** Note that the calculation of yield efficiency is inverted for endergonic cases, thus it cannot be directly compared with the exergonic cases.

| Inorganic Carbon Concentration (mol/L) | $\Delta G_{overall}$ (kJ/$mol_{Ac\text{-}CoA}$) | Minimum Net ATP Required (mol ATP/$mol_{Ac\text{-}CoA}$) | Minimum Net Energy Required (kJ/mol) | Energy Utilization Efficiency (% $kJ_{required}$ / $kJ_{spent}$) | Maximum MDF (kJ/mol) |
|---|---|---|---|---|---|
| 0.1 | -21.89 | -0.1 | -5 | N/A | 0.276 |
| 0.01 | -10.47 | 0.2 | 10 | N/A | 0.573 |
| 0.001 | 0.95 | 0.5 | 25 | 3.78 | 0.848 |
| 0.0001 | 12.36 | 0.8 | 40 | 30.9 | 1.124 |

mol acetyl CoA synthesized for the $C_{ic}$ range studied, or the equivalent of spending 40 to -5 kJ/ mol of free energy per mol acetyl CoA synthesized. Note that even when the reaction is slightly exergonic (for the case of $C_{ic}$ = 0.01 mol/L in Table 4), the cell still needs to spend energy to achieve $CO_2$ fixation. Thus, it can be strongly inferred that the majority of the ATP energy spent in the reverse TCA cycle is used to increase driving forces. These results illustrate how organisms can overspend ATP in order to create large driving forces in this pathway [15]. As an example, the citrate to oxaloacetate reaction would be very likely to proceed in the reverse direction without the investment of one ATP in the form of SLP, as the concentration of its reactant is at the minimum bound and its product at the maximum bound as seen in Fig 6.

Further observation of the optimised configurations of the individual variants reveals that only four to six (depending on $C_{ic}$) of the 46 variants can achieve the maximum MDF listed in Table 4. These variants share two common features: 1) ferredoxin is the electron carrier for the reduction of succinyl CoA to 2-ketoglutarate, and 2) all the remaining electron carriers involved are either NAD or NADP. This could be an indication that the microbes that have evolved to utilize quinones or FAD for these reduction reactions are energetically sub-optimal at these environmental conditions and are doing those for other reasons such as the availability of these carriers at high energy from other metabolic reactions.

In Fig 7, the Pareto curve generated via the multi-objective optimisation is presented for different external concentrations of inorganic carbon, and the most optimal pathway variant is presented at each data point. In this case, however, the horizontal axis represents the net energy spent (instead of the net energy recovered) in the pathway in units of ATP equivalents. The upward-sloping curves are to be expected; as more energy in the form of proton transloca- tions is spent into the pathway, it is possible to run the individual reactions with higher driving forces. The leftmost point of each curve shows the absolute limit of energy efficiency for the pathway, representing the maximum MDF at minimum energy expenditure. Since that point

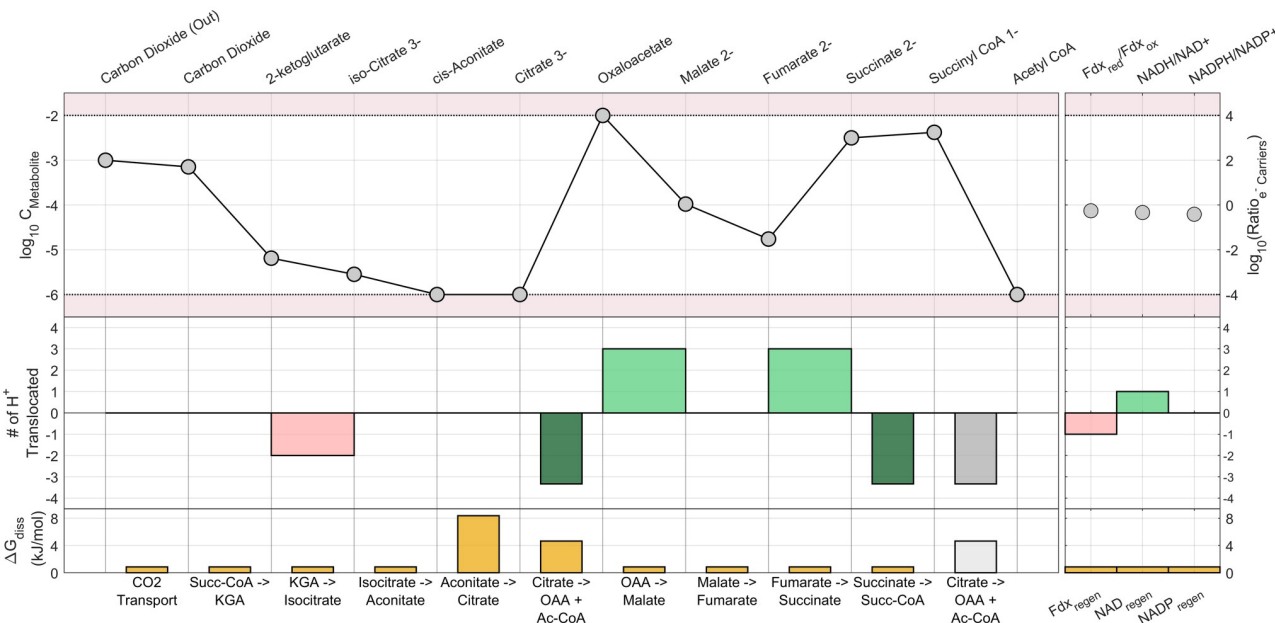

**Fig 6. Detailed bioenergetic breakdown of the reverse TCA pathway, evaluated at the parameters from Table 4, and $C_{ic}$ = 0.001 mol/L.** The wide red bars represent proton translocations invested and the bright green bars represent proton translocations recovered. The narrow dark green bars represent SLP. A greyed-out bar indicates a reaction that's repeated for the sequential listing of species. The species are arranged in the order they react and are produced by the pathway, allowing for visual identification of bottlenecks.

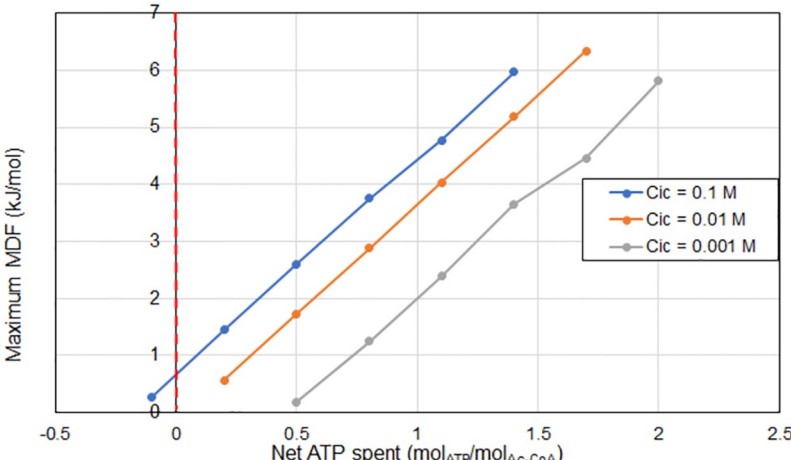

**Fig 7. Pareto curves for the reverse TCA cycle evaluated at the same parameters as Table 4 excluding $C_{ic}$.** The horizontal axis represents the net energy spent in the pathway in units of proton translocations, and the vertical axis represents the maximum MDF possible for the pathway with that energy recovery. The various points are obtained by changing the value of epsilon (number of translocations conceded) from 0 to 5.

is to the right of the y-axis for the $C_{ic}$ = 0.01 and 0.001 mol/L scenarios, it is clear that it is not possible to fix $CO_2$ using the reverse TCA cycle under these conditions without an energy investment. However, for the $C_{ic}$ = 0.1 mol/L scenario, which could be achieved in a bioreactor, theoretically it appears energetically possible to fix $CO_2$ exergonically via the reverse TCA pathway. Although possible, this would imply very low driving forces and maximum MDF, and a flux-force efficacy of only 6% (calculated using the formula in section 1).

## 4. Conclusion

The methodology for pathway analysis developed in this work allows for the mechanistic elucidation of the most energetically efficient among candidate microbial metabolic pathway variants under different conditions. The method solved uniquely the optimum metabolic intermediate concentrations for maximum energy recovery and maximum limiting driving force. The methodology also successfully allowed direct comparison of pathway variants, in terms of their energy recovery efficiency and potentially rate-limiting driving force. Given its characteristics and linearity as well as the optimisation techniques used, the method is computationally inexpensive and provides high confidence in global optimality for each pathway variant evaluated. This makes the method fast, robust and highly scalable for larger networks while maintaining a high degree of flexibility to accommodate a large range of additional constraints. Most importantly, since the approach is based entirely on thermodynamic principles, the pathway analysis results were obtained with minimum need for assumptions and/or parameter estimations.

The results obtained for the two types of pathways evaluated are consistent with the mechanistic descriptions and expected efficiencies and bottlenecks in those pathways, as well as the reported literature. The trade-off between energy efficiency and driving force (rate) is mechanistically computed using the method and can be visualized in the Pareto fronts shown in the two example pathways. In addition, the sensitivity of the pathways to environmental conditions was presented and discussed. Finally, since the methodology is developed in the language of the well-studied mathematical field of optimisation, one can introduce a plethora of

mathematical concepts to improve our analysis as required by the particular modelling objectives. For example, the notion of irreducible inconsistent subsets can be used to detect particular reactions or metabolite concentrations that are blocking infeasible pathways, while the concept of slack allows the identification of yield-limiting steps in optimised pathways.

Through the application of the methodology to six metabolic pathways for propionate oxidation it was found that the pathways via lactate and hydroxypropionyl CoA (both linear and cyclical) yielded the highest ATP recovery under the conditions evaluated. The via lactate and cyclical via hydroxypropionyl pathways also had the highest MDF under that set of conditions, and consistently remained so as energy recovery was sacrificed to increase driving forces.

The reverse TCA cycle, an anabolic pathway for microbial carbon fixation was analysed using the methodology under different concentrations of inorganic carbon. The analysis shows that although it appears possible to fix carbon without spending cellular energy overall, the very small driving forces would imply likely an unfeasibly slow kinetic rate. In addition, based on the analysis, it appears that variants that use ferredoxin rather than NAD(P) for the reduction of succinyl-CoA to ketoglutarate, as well as NAD(P) instead of FAD or quinone for the reduction of oxaloacetate to malate and fumarate to succinate were capable of achieving higher MDF values. This suggests that the utilization of the other carriers for those reactions may be driven by other factors than purely thermodynamic ones.

It is important to note that other factors apart from thermodynamics influence the rates of reactions in a pathway. These include the regulation of enzyme activity by metabolites that are either before or after the reaction catalyzed by the enzyme. Although these mechanisms of regulating reaction rates must still adhere to the limits imposed by thermodynamic driving forces, they play a crucial role in altering the rates of individual reaction steps in the pathway. Comprehensive dynamic pathway models that describe enzyme kinetics, which are not targeted in this study, should take into account enzyme control and other phenomena that impact reaction rates.

## Supporting information

**S1 Text. Supplementary material text containing tables A-C and Fig A.**
(PDF)

## Author Contributions

**Conceptualization:** Ahmed Taha, Jorge Rodríguez.

**Formal analysis:** Ahmed Taha.

**Funding acquisition:** Jorge Rodríguez.

**Methodology:** Ahmed Taha, Mauricio Patón, Julio R. Banga, Jorge Rodríguez.

**Project administration:** Jorge Rodríguez.

**Resources:** Jorge Rodríguez.

**Software:** Ahmed Taha, David R. Penas, Julio R. Banga.

**Supervision:** Julio R. Banga, Jorge Rodríguez.

**Validation:** Mauricio Patón.

**Visualization:** Ahmed Taha.

**Writing – original draft:** Ahmed Taha.

**Writing – review & editing:** Mauricio Patón, David R. Penas, Julio R. Banga, Jorge Rodríguez.

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
