## [Decision Letter · Decision Letter 0]

31 Mar 2023

Dear Dr. Rodríguez,

Thank you very much for submitting your manuscript "Optimal evaluation of  energy yield and driving force in microbial metabolic pathway variants" for consideration at PLOS Computational Biology.

As with all papers reviewed by the journal, your manuscript was reviewed by members of the editorial board and by several independent reviewers. In light of the reviews (below this email), we would like to invite the resubmission of a significantly-revised version that takes into account the reviewers' comments.

Two experts in thermodynamics have reviewed your submission. Based on their feedback, major revisions are required. You will see, however, that the requested revisions are not burdensome but I do ask that you take them seriously as it is important to respond to these in a diligent manner. One comment is in regard to characterizing the minimum driving force using shadow prices. While it would be fine to include such an analysis, I believe that it is only necessary to mention that this is possible and give references; using thermodynamics to characterize driving forces is always preferable to borrowing economic principles.

We cannot make any decision about publication until we have seen the revised manuscript and your response to the reviewers' comments. Your revised manuscript is also likely to be sent to reviewers for further evaluation.

Sincerely,

William Cannon

Guest Editor

PLOS Computational Biology

Mark Alber

Section Editor

PLOS Computational Biology

Dear Dr. Rodríguez,

Two reviewers have now reviewed your submission. Based on their feedback, major revisions are required. You will see, however, that the requested revisions are not burdensome but I do ask that you take them seriously as it is important to respond to these in a diligent manner. One comment is in regard to characterizing the minimum driving force using shadow prices. While it would be fine to include such an analysis, I believe that it is only necessary to mention that this is possible and give references; using thermodynamics to characterize driving forces is always preferable to borrowing economic principles.

We look forward to your revisions.

Best regards,

Bill Cannon

Reviewer's Responses to Questions

**Comments to the Authors:**

Reviewer #1: The review is uploaded as an attachment.

Reviewer #2: Taha et al. present a methodology for designing and evaluating metabolic pathways, with a special focus on energy yields in the form of ATP and proton translocations. They show that different pathways exhibit different trade-offs between this yield and the thermodynamic efficiency in terms of driving forces, and demonstrate, using a Pareto analysis, how this is especially relevant in cases where energy is scarce such as anaerobic fermentation and the reverse TCA cycle. This work is a useful extension of previous pathway analysis methods based on thermodynamics.

* In general, it is not completely clear why the authors treat alternative electron carriers so differently than proton translocations. In both cases, not all combinations exist in nature but could potentially be engineered or evolved. Both are discrete choices. Similar to changing the proton stoichiometry, switching from NAD+ to a quinone could boost the driving force at the expense of the amount of energy conserved as ATP. And, finally, alternative carriers can also be dealt with using integer variables - using reaction indicators and adding a constraint forcing that only one can be active at a time.

Furthermore, in the results regarding the rTCA cycle, it seems that this equivalence was overlooked by the authors. They state that using more energetic electron donors (ferredoxin instead of NAD and NAD instead of quinone) yield better MDFs solutions, which should not come as a surprise. However, cells might prefer using less energetic donors in order to conserve more energy in ATP (or rather spend less of it). Including the potential ATP yield as they did for proton translocations would make the analysis much more valuable.

* Previous versions of MDF used shadow prices to identify and quantify the reactions which constrain the driving forces in the pathway. It would be helpful to add that information to the analyzed pathways in the results section. These reactions should also be the ones where adding extra driving force via SLP or proton translocation would give the largest benefit. Or, if that is not the case, it could highlight the advantage of this current method.

* Although the constraints used in the optimization process are written out explicitly and clearly, their justification is often lacking:

- Line 278: What is the justification for having a non-zero threshold for ΔGr? And why is this necessary given the fact that MDF solutions are anyway far from 0 for all reactions. Also, this constraint does not appear in Table 1 (summary of the LP). Is it converted to "F_i" constants? How are they chosen? If not, where do they come from in the "First Optimization"?

- Lines 317-328: The argument that proton translocation energy cannot exceed SLP is unclear and there is no reference given. Also the claim that leaving it unbounded increases computational time seems reasonable but no quantitative evidence is given and it would be helpful to show that the end results are similar at least for one of the examples.

- Lines 335-338: Why is it more reasonable to bound the total pool of a conserved moiety? The text states this as an obvious fact without a real explanation. Indeed, the total pool cannot be changed by the pathway itself, but it can definitely be changed by reactions that are not in the pathway and therefore it could make sense to allow it in the optimization step.

Minor comments:

* Line 84: It's not clear to me whether ΔG_diss is the same as the change in Gibbs free energy or minus that. In the equation it seems to be positive, which makes sense, but later in line 88 the driving force is equated to (-ΔG_diss) which implies the opposite.

* Line 137: "terminal electron acceptor for an oxidative pathway is hydrogen". Hydrogen cannot accept any more electrons. Perhaps "electron donor" was meant here?

* Line 194: "in kJ/mol.K" - replace "." with "/"

Equation 16: applying an upper bound (but not a lower bound, unfortunately) on a log_sum_exp() function is a convex constraint, perhaps that could be useful here?

* Line 359-361: "no driving force forced energy dissipation was required" -> unclear sentence

* Line 441: Shouldn't the MDF value be positive (i.e. +1.5 kJ/mol)?

**Have the authors made all data and (if applicable) computational code underlying the findings in their manuscript fully available?**

Reviewer #1: Yes

Reviewer #2: Yes

PLOS authors have the option to publish the peer review history of their article (what does this mean?). If published, this will include your full peer review and any attached files.

Reviewer #1: No

Reviewer #2: No
---

## [Decision Letter · Decision Letter 1]

24 May 2023

Dear Dr. Rodríguez,

Thank you very much for submitting your manuscript "Optimal evaluation of  energy yield and driving force in microbial metabolic pathway variants" for consideration at PLOS Computational Biology. As with all papers reviewed by the journal, your manuscript was reviewed by members of the editorial board and by several independent reviewers. The reviewers appreciated the attention to an important topic. Based on the reviews, we are likely to accept this manuscript for publication, providing that you modify the manuscript according to the review recommendations.

Sincerely,

William Cannon

Guest Editor

PLOS Computational Biology

Mark Alber

Section Editor

PLOS Computational Biology

Reviewer's Responses to Questions

**Comments to the Authors:**

Reviewer #1: The review is uploaded as an attachment.

Reviewer #2: In general, the authors answered the majority of my and the other reviewer's comments.

However, a few things could be improved still:

- Regarding the iteration of electron carriers: Indeed, adding extra binary (or integer) variables to an MILP will increase computational time and might (in the worst case) be as slow as looping through the entire combinatorial space (i.e. exhaustively). However, this argument is just as relevant for alternative electron carriers as it is for various proton translocation stoichiometries. If, indeed, no benefit is provided by using reaction indicators, then one can simply give up on integer variables completely and run a series of LPs (iterating the possible proton stoichiometries). This will also have the added benefit of being able to analyze the sensitivity of the MDF to the different reaction energies and metabolite concentration bounds (see my last comment).

- The upper bound on H+ translocations (based on SLP) is reasonable, but could be explained and discussed more clearly. Relaxing it seems feasible, at least for some examples, and then one could see whether the results change significantly. If not, that is another reason to keep it (saving computation time). If they do, it could lead to an interesting discussion (regarding tradeoffs in evolution). Generally, I agree that if there is enough energy for SLP - that should be preferred by evolution. But it doesn't seem like a strict rule and exceptions to it might exist in nature already.

- Regarding terms and notations, confusion still remains in several places. For example, in line 289 the authors write "When ∆Gr is negative, that energy is dissipated and is usually referred to as the reaction's thermodynamic driving force." - this phrasing might be confusing (does it suggest that when ∆Gr is positive something different happens?). Later, ∆Gmin is defined as the minimum required driving force but based on equation (7) it should probably be negative as it is an upper bound on ∆Gr (if it were positive that would be redundant, since ∆Gr < 0 anyway). Is ∆Gmin the optimization parameter that is later denoted B (although B is positive)? If so, the sentence "A threshold minimum required driving force (ΔGmin) *is necessary* for any reaction to proceed" is quite confusing, as it suggests a physical reason rather than the kinetic motivation behind the MDF method. Furthermore, isn't B missing from the thermodynamic constraints in Table 1 (otherwise it doesn't seem to have any effect on the MILP)?

- Finally, regarding identifying bottlenecks, indeed MILP solvers do not provide shadow prices and therefore it is not as straightforward to see which constraints are "active". However, it seems to me that this information should still be possible to get (and the authors suggest a few ways in the discussion). Perhaps after finding the optimal H+ translocation stoichiometry, one could run a standard LP only for (re-)optimizing the concentrations - the optimum should probably be at the same place and the shadow prices would be available then. This is quite a significant omission from the analysis and if it requires little effort to achieve, could be quite informative.

- The code is indeed freely available on Zenodo. However, there is no documentation or instructions on how to use it. It is ridden with commented out lines that were probably relevant during the study but render it unapproachable to others. "main_nl.m" refers to a non-existing .xlsx file. Please ensure that a fresh download of the repository can be run by other users, clear out the comments, and add a proper README.

**Have the authors made all data and (if applicable) computational code underlying the findings in their manuscript fully available?**

Reviewer #1: None

Reviewer #2: Yes

PLOS authors have the option to publish the peer review history of their article (what does this mean?). If published, this will include your full peer review and any attached files.

Reviewer #1: No

Reviewer #2: No

Figure Files:

Data Requirements:

Reproducibility:

References:

---

## [Editor Report · Decision Letter 2]

12 Jun 2023

Dear Dr. Rodríguez,

We are pleased to inform you that your manuscript 'Optimal evaluation of  energy yield and driving force in microbial metabolic pathway variants' has been provisionally accepted for publication in PLOS Computational Biology.

Best regards,

William Cannon

Guest Editor

PLOS Computational Biology

Mark Alber

Section Editor

PLOS Computational Biology

---

## [Editor Report · Acceptance letter]

26 Jun 2023

PCOMPBIOL-D-23-00252R2 

Optimal evaluation of  energy yield and driving force in microbial metabolic pathway variants

Dear Dr Rodríguez,

I am pleased to inform you that your manuscript has been formally accepted for publication in PLOS Computational Biology. Your manuscript is now with our production department and you will be notified of the publication date in due course.

With kind regards,

Zsofia Freund
